# Young Patients with Anorexia Nervosa: The Contribution of Post-Traumatic Stress Disorder and Traumatic Events

**DOI:** 10.3390/medicina57010002

**Published:** 2020-12-22

**Authors:** Paola Longo, Enrica Marzola, Carlotta De Bacco, Matilde Demarchi, Giovanni Abbate-Daga

**Affiliations:** Eating Disorders Center, Department of Neuroscience, University of Turin, 10126 Turin, Italy; paola.longo@unito.it (P.L.); enrica.marzola@unito.it (E.M.); carlideb@yahoo.it (C.D.B.); matilde.demarchi@gmail.com (M.D.)

**Keywords:** eating disorders, anorexia nervosa, risk factors, trauma, post-traumatic stress disorder

## Abstract

*Background and Objectives*: Anorexia nervosa (AN) is a complex disorder whose etiopathogenesis involves both biological and environmental factors. The aims of the present study were to retrospectively analyze risk factors in young patients with AN and to assess differences in clinical and eating-related symptoms between patients with and without a diagnosis of post-traumatic stress disorder (PTSD) and with or without a history of acknowledged risk factors. *Materials and Methods*: Sixty-four patients with AN (<25 years old) were recruited and completed an anamnestic evaluation and the following self-report measures: Eating Disorder Examination Questionnaire (EDE-Q), Childhood Trauma Questionnaire (CTQ), State-Trait Anxiety Inventory (STAI-Y), Beck Depression Inventory (BDI), Life Events Checklist (LEC), and Dissociative Experience Scale (DES). The PTSD diagnosis was assigned according to the Structured Clinical Interview for the DSM-5 (SCID-5). *Results*: The most frequent risk factors were those associated with relational traumatic events and familiarity for psychiatric disorders. Higher severity of body-related symptoms (i.e., those symptoms impacting on body image and perception and leading to body concerns) emerged in patients with PTSD, versus patients without PTSD diagnosis; however, after controlling for dissociative symptoms, only differences in BMI remained significant. Concerning other risk factors, those with a history of childhood trauma were more depressed than patients without such history and those with familiarity with eating disorders reported more AN-related hospitalizations in the past than those individuals without familiarity. *Conclusion*: These results suggest the importance of investigating the presence of risk factors and PTSD diagnosis in patients with AN, and to treat post-traumatic symptoms in young patients in order to decrease the risk of developing severe forms of AN. Moreover, a particular focus on those patients with a family member affected by an eating disorder could be of clinical utility.

## 1. Introduction

Mental disorders are complex and severe illnesses often leading to psychosocial and physical disability [1]. In the last decades, many studies focused on the recognition of potential risk factors (i.e., measurable features that anticipate the onset of a disorder [2]), generally agreeing on the role of both biological and environmental aspects in the development of mental diseases [3,4,5]. From a biological standpoint, a genetic basis has been described for the majority of psychiatric disorders, as suggested also by the presence of familiarity traits (i.e., presence of psychopathological symptoms in patients’ first or second-degree relatives), for example in mood disorders [6,7]. As regards environmental factors, an adverse family context plays an important role with research showing that a low family income increases the risk of developing an anxiety disorder [8], and family discord and the lack of family support and cohesion are related to higher odds of psychopathology in adolescence [3,9]. Other risk factors commonly considered in psychiatry are represented by alcohol and substance abuse; for example, alcohol use in adolescence was described as a predictor of borderline personality disorder, while cannabis use was correlated to an earlier onset of schizophrenia [10,11]. However, the correlation between substance and alcohol abuse and psychiatric disorders (e.g., anxiety disorders, eating disorders) is less studied. An important role as a psychological/environmental risk factor is also played by childhood adversities, maltreatment, and lifetime stressful events, as has been demonstrated for eating disorders [12,13,14] (EDs).

EDs are debilitating illnesses characterized by abnormal eating attitudes and psychosocial and physical impairment. As described for other mental disorders, the etiopathogenesis of EDs is represented by an interaction between biological and environmental factors. As regards biological features, research showed that genetic aspects contribute to the etiology of EDs [15,16]. In this vein, family studies described a higher prevalence of EDs in relatives of patients with EDs, showing even a greater contribution of genetic factors, compared to environmental ones, in familiarity [17,18,19,20]. Concerning environmental aspects, as stated above, stressful and traumatic events are considered EDs risk factors [21]. Indeed, individuals with EDs are more likely to report a history of trauma, and individuals who have suffered a traumatic event are more likely to develop maladaptive eating behaviors [22]. Furthermore, childhood abuse was described as a non-specific risk factor for EDs [23,24]: in particular, the incidence of bulimic syndromes was estimated as 2.5 times higher in patients reporting an episode of childhood sexual abuse, with an increasing risk in case of multiple episodes [23]; however, the impact of a Post-Traumatic Stress Disorder (PTSD) full-blown diagnosis on Anorexia Nervosa-related symptoms is to date less studied. Concerning alcohol and substance abuse, Substance Use Disorder has been described as a risk factor for the development of anorexia nervosa (AN) [25]. Moreover, the literature reported shared genetic mechanisms between vulnerability to EDs and to substance use disorder, with a higher correlation between bulimic symptoms and alcohol use and substance dependence [26,27]. However, studies on the role of alcohol and substance abuse in EDs are few, and generally, there is a lack of literature investigating extensive risk factors in young (i.e., <25 years old) patients with EDs recruited during the first years of illness. Therefore, some gaps in the literature need to be noted: few studies investigated substance and alcohol abuse as risk factor for AN as well as young patients at the onset of their illness; still, to the authors’ knowledge, very few studies focused on both familial and environmental risk factors. Moreover, Solmi et al., (2020) with an umbrella review accounted for a lack of well-established risk factors for EDs; in fact, even considering several risk factors (e.g., childhood sexual abuse, physical abuse, substance use, impulsivity), strong evidence was not found for any ED, particularly AN [28].

Bearing these gaps in mind, we aimed to deepen the issue of risk factors in AN; therefore, our first goal was to retrospectively identify and describe risk factors in young patients with AN focusing on childhood abuse, traumatic events (discriminating between events occurred during the life-span and those specifically occurred before the onset of AN, and between relational and non-relational traumas), alcohol/substance abuse, and familiarity for general and eating-related psychopathology. As a second goal, we aimed to assess differences in clinical and eating-related symptoms, and in general psychopathology (e.g., anxiety and depression) between patients with and without a diagnosis of PTSD and with or without a history of the aforementioned risk factors. Concerning the first goal, we expected to find a higher frequency of certain (e.g., trauma-related) risk factors; secondly, we hypothesized to observe a more severe clinical presentation in patients with AN and comorbid PTSD when compared to those with AN but without PTSD.

## 2. Materials and Methods

### 2.1. Participants

A sample of 64 inpatients with AN, both restricting (RAN) and binge-eating/purging (BPAN) subtypes, was recruited at the Eating Disorders Centre of the ‘Città della Salute e della Scienza’ hospital of the University of Turin, Italy.

Inclusion criteria were: (a) maximum age 25 y.o.; (b) formal diagnosis of AN according to DSM 5 criteria as assessed by an experienced psychiatrist per clinical interview [29]; (c) female gender. The following exclusion criteria were applied: (a) Wechsler Adult Intelligence Scale-Revised Intellectual Quotient score < 85 [30]; (b) medical problems such as diabetes or epilepsy; (c) anamnesis of cranial trauma with loss of consciousness; (d) comorbid psychotic spectrum disorders and/or bipolar disorders; (e) substance and/or alcohol abuse.

All the participants provided written informed consent according to the ethical committee of the University of Turin. The present study was approved by the ethical committee under the registration number CS2/840 (approval date: 18 June 2018).

### 2.2. Procedure

All participants were interviewed upon admission by an experienced psychiatrist to confirm the diagnosis of AN and collect clinical and demographic data. At the same time point, trained nurses measured participants’ body mass index (BMI). During the first days of hospitalization, patients completed a standardized diagnostic interview and several self-report questionnaires.

### 2.3. Materials

Participants were administered the Structured Clinical Interview for the DSM-5 (SCID-5) [31] to assess the presence of a current diagnosis of PTSD. Moreover, patients completed the following measures:Eating Disorder Examination Questionnaire (EDE-Q) [32,33]: This tool assesses the typical attitudes and behaviors of eating disorders as they occurred during the last 28 days. It consists of 28 items and four subscales: dietary restraint, eating concerns, weight concerns, and shape concerns. A global score is also provided. Internal consistency is acceptable with Cronbach’s alpha values ≥0.90 [33].Childhood Trauma Questionnaire (CTQ) [34]: The questionnaire is composed of 28 items measured on a five-point Likert scale ranging from 1 (never true) to 5 (very often true). The severity of five types of childhood abuse is assessed and reported by the following subscales: childhood emotional abuse, childhood physical abuse, childhood sexual abuse, childhood emotional neglect, childhood physical neglect, and global score. Cut-off values for each scale are also available for categorical analysis. Internal consistency is good, as estimated by a Cronbach’s alpha value of 0.91 [35].State-Trait Anxiety Inventory (STAI-Y) [36]: The measure consists of two sets of questions: 20 questions on the current state of anxiety to assess state anxiety, whereas the other 20 questions investigate the trait anxiety as a stable trait. Answers range on a scale from 1 (never) to 4 (always). Internal consistency values range from 0.86 to 0.95 [36].Beck Depression Inventory (BDI) [37]: The 21-items questionnaire investigates the severity of depressive symptoms, as described by the global score. Indeed, a score from 0 to 4 reflects low depression, from 5 to 15 moderate symptoms, and from 16 to 39 severe depressive symptomatology. Internal consistency is high, as indicated by a Cronbach’s alpha value of 0.86 [38].Life Events Checklist (LEC) [39]: It is a self-report trauma assessment of the Clinician-Administered PTSD Scale. LEC is used to investigate the occurrence, type, and timing of lifetime potentially traumatic events listing 16 types of traumatic events: natural disaster, accidents or explosion, car or train or flight accident, severe accident at job place or home, exposure to toxic substances, physical violence, being assaulted with a weapon, sexual assault, other kinds of sexual assault, being kidnapped, exposure to fighting, physical illness, severe human suffering, sudden or violent death, unexpected close person’s death, serious harm or death of someone close to her, any others stressful or traumatic events. Subjects are asked to address what events they have suffered, if they were the victim, the witness, or close to a victim of the traumatic event, and at what age the event occurred.Dissociative Experience Scale (DES-II) [40]: It is a self-report questionnaire assessing the presence of several dissociative symptoms such as absorption, amnesia, derealization and depersonalization. It consists of 28 items asking the patients how frequently the symptoms occur on a scale ranging from 0 (never) to 100 (always). A total score above 30 suggests the presence of a dissociative disorder. The internal consistence of the Italian version is good (Cronbach’s alpha: 0.81–0.94 [41]).

### 2.4. Statistical Analysis

The SPSS 26.0 statistical software package (IBM SPSS Statistics for Windows, Version 26.0. Armonk, NY, USA: IBM Corp) was used for data analysis. Descriptive analyses were conducted to assess the distribution of clinical and demographic variables and risk factors. As the distribution of the groups was not normal, a Wilcoxon-Mann-Whitney test was run to investigate differences between groups as regards continuous variables. For differences between groups with categorical variables, Exact Fisher’s Test was used instead. Finally, analysis of covariance (ANCOVA) was conducted to control for possible confounding variables.

## 3. Results

### 3.1. Demographic and Clinical Features of the Sample

The sample consisted of 64 female inpatients with AN, 43 with RAN, and 21 with BPAN. Patients’ mean age was 20.2 y.o. (SD = ± 2.3). As regards demographic variables, 2 patients were employed (3.3%), 51 (83.6%) were students, while 8 (13.1%) were unemployed. The majority of the sample (81.4%) lived with their family of origin. See Table 1 for clinical variables. Moreover, 11 patients (17.2%) practiced self-harm, 10 (15.6%) reported a suicide attempt, and 2 (3.1%) reported previous abortions.

### 3.2. Risk Factors

The most frequently reported risk factors were psychiatric familiarity and childhood traumatic events (Table 2).

As regards trauma-related risk factors, the most frequent events were lifetime traumatic events, mostly relational trauma, and traumatic events before AN onset (Table 3). In particular, concerning relational traumas, 37.7% of patients experienced physical violence, 8.2% were assaulted with a weapon, 9.8% suffered sexual assault and 23% other kinds of sexual assault, and 4.9% reported to have been kidnapped.

### 3.3. Differences between Patients with and without PTSD and Risk Factors

Patients with PTSD comorbidity had a higher BMI than non-PTSD patients. Furthermore, patients with a PTSD diagnosis reported significantly higher scores in shape concern, weight concern, and total score of the EDE-Q (but not restraint and eating concern scales), trait anxiety, and dissociation when compared to those without PTSD. Since relational traumatic events lead often to dissociative symptoms (Longo et al., 2020), we controlled our findings for this covariate as well. Therefore, after controlling for dissociative symptomatology, only the difference in BMI remained significant (Table 4). Moreover, patients with BPAN were diagnosed with a comorbid PTSD more frequently than those with RAN (Fisher’s exact test: *p* = 0.008), while no differences between PTSD and non-PTSD groups were shown in frequency of self-harm (Fisher’s exact test: *p* = 1.000) and suicide attempts (Fisher’s exact test: *p* = 0.213).

Concerning other risk factors (i.e., childhood trauma, trauma before AN onset, multiple traumas, psychiatric familiarity, familiarity for EDs), few significant results emerged: BDI and DES scores were higher, and BPAN diagnosis was more frequent in patients with childhood trauma than in those without such a history. Furthermore, BMI was higher in the group reporting multiple traumas than in the non-multiple trauma group; finally, patients with familiarity for EDs reported a higher number of hospitalizations when compared to those without such a familiarity (for details see Appendix A).

## 4. Discussion

The present study retrospectively described risk factors in young patients with AN aiming then to evaluate the impact of presence versus absence of both a comorbid diagnosis of PTSD and of risk factors on young patients’ clinical presentation. There were two main findings that emerged: first, psychiatric familiarity and childhood traumatic events were the most frequently reported risk factors while, concerning trauma-related risk factors, the most frequent events were lifetime traumatic events, mostly relational trauma, and traumatic events before AN onset. In contrast, alcohol and substance use was scarcely reported as a risk factor in patients with AN. Second, overall clinical severity was not different between patients with and without trauma history, while patients with AN and comorbid PTSD showed more severe body-related symptoms (e.g., shape and weight concerns) than patients with AN but without PTSD, thus partially confirming our a priori hypotheses.

Descriptive analysis showed that the most frequently reported risk factors among young patients with AN were familiarity with psychiatric disorders and those associated with traumatic events, especially relational trauma and childhood abuse, while the frequency of alcohol and substance abuse was low. Taken together, our results are in line with studies supporting the familial contribution to the etiology of AN [15,16] and the role of traumatic events and childhood abuse, in particular emotional and physical neglect, as risk factors for EDs [22,23,24,42,43]. Moreover, both having suffered a trauma and having grown up in a family with a member affected by a psychiatric illness could lead to a series of further risk factors for mental diseases. For instance, negative affectivity, often a consequence of a history of abuse, was described to be a risk factor for the development of body concern [12]; furthermore, the lack of family cohesion and support during adolescence tends to increase the risk for psychiatric symptoms in general [3,9]. As regards alcohol and substance use, the present study suggests a limited role of psychoactive substances in the development of AN; this result is not in line with previous literature describing EDs and substance use disorders as frequently co-occurring [27], and a role of substance use disorder as a risk factor for AN [25]. However, it is of note that the correlation between alcohol/substance use and EDs is higher among individuals with bulimic symptoms [44], while the present study focused on AN, mostly RAN. However, existing data on the topic are few and further longitudinal studies are needed to investigate this hypothesis.

Patients with and without a comorbid PTSD diagnosis showed several differences in their clinical presentation, mostly on body-related symptoms. Therefore, a full-blown diagnosis of PTSD—rather than the mere presence of trauma history—should be carefully investigated in clinical practice, also in the light of the literature describing PTSD symptoms as central in maintaining and strengthening AN [22]. Unexpectedly, a higher BMI was observed in patients with PTSD compared with those without this diagnosis therefore partially disconfirming earlier literature [22,45]. However, a couple of caveats should be considered: first, BMI could not be an effective proxy for AN severity, as already proposed [46,47]; second, a higher BMI could be also related to the greater presence of patients with BPAN in the PTSD group. In fact, our data are in line with earlier literature reporting that patients with BPAN are more likely to report trauma history and post-traumatic symptoms compared to patients with RAN [14,48]. It is also noteworthy that a diagnosis of PTSD in comorbidity with AN could contribute to a potentially different clinical presentation—also impacting on weight—when compared to patients with a pure diagnosis of AN. Patients with PTSD showed higher scores in shape concern, weight concern, and total score of EDE-Q, in keeping with previous findings on post-traumatic symptoms and AN severity [14,22]. These results also highlight that patients with trauma history and a PTSD diagnosis reported more severe body-related (e.g., body concern) rather than eating-related symptoms as measured by EDE-Q. This datum is interesting since it is known that the body is seriously affected by abuse [49,50]. Interestingly, anxiety traits were higher in patients with AN and PTSD compared to patients without PTSD, in line with previous literature describing a higher frequency of PTSD diagnosis in individuals predisposed to anxiety and stress-related psychopathology [51]. However, when dissociation was added as a covariate to the model, all differences but BMI lost significance. Therefore, the hypothesis could be raised that post-traumatic dissociation could be the main symptomatology cluster involved in the severity of eating-related symptoms in patients with AN and PTSD. Indeed, previous literature described a role of dissociation as a mediator between traumatic events and the onset of EDs, and an impact of dissociative symptoms on the severity of eating-related symptoms [52].

As regards other risk factors, results showed higher levels of dissociation and depression in patients with childhood trauma compared to patients without a history of such abuse. These data are in line with Toyoshima and colleagues (2020) [53] that demonstrated a direct effect of childhood maltreatment on depressive symptoms, and with some studies showing higher levels of dissociation in patients with EDs and a history of childhood abuse compared to patients without childhood trauma [54,55]. Finally, patients with familiarity for EDs reported a significantly higher number of hospitalizations compared to patients without this risk factor. It could be speculated that patients with a family member affected by EDs could develop a more complex and severe form of AN because of the interaction between genetic predisposition and environmental influences. Moreover, parents with a history of EDs could have difficulties in managing their children’s illness, leading thus to the need for hospitalization. Despite some strengths including the focus on young patients and the analysis of several risk factors, some limitations should be acknowledged as well: the sample size was relatively small; the design was cross-sectional, so causal links cannot be fully clarified; the two subtypes of AN were not considered; risk factors related to socioeconomic status and personality features were not taken into account; finally, a healthy control group was not recruited, thus comparisons with healthy subjects are not available.

## 5. Conclusions

To our knowledge, this is the first study evaluating numerous and different risk factors in young patients (<25 years old) with AN thus focusing on the first stages of the illness. The present study described relational and childhood trauma, and familiarity for psychiatric disorders as the most common risk factors in young patients with AN. Furthermore, the co-diagnosis of PTSD and AN resulted in an increase in clinical severity, especially for body-related symptoms, whereas history of traumatic events was not a specifier of severity. In this context, future studies could focus on biological mechanisms related to PTSD such as cortisol reactivity, already known to be altered in patients with AN [56], and epigenetic regulation [57]. Taken together, these results have clinical implications suggesting the importance of properly detecting risk factors in young patients with AN and to target treating PTSD symptoms in order to decrease the risk of developing severe forms of AN. Moreover, support and prevention programs for adolescents with a relative with a psychiatric disorder could be of clinical utility.

## Figures and Tables

**Table 1 medicina-57-00002-t001:** Clinical variables of inpatients with anorexia nervosa (AN).

	Mean (SD)
BMI	15.2 (2.6)
Duration of illness, years	3.4 (2.7)
Age at onset, years	16.8 (2.4)
N of previous AN-related hospitalizations	1.9 (1.5)

BMI = body mass index.

**Table 2 medicina-57-00002-t002:** Risk factors before AN onset in young patients with AN.

	Inpatients with AN (n = 64)	
	Yes n (%)	No n (%)
Psychiatric familiarity	24 (41.4)	34 (58.6)
EDs familiarity	10 (16.7)	50 (83.3)
Childhood traumatic event *	51 (81)	12 (19)
Substance use	3 (4.8)	59 (95.2)
Alcohol use	1 (1.6)	61 (95.3)

EDs = eating disorders; * = person with at least one of total scores of Childhood Trauma Questionnaire over cut off.

**Table 3 medicina-57-00002-t003:** Trauma-related risk factors in young patients with AN as measured by the Life Events Checklist (LEC) and Structured Clinical Interview for the DSM-5.

	Inpatients with AN (n = 64)	
	Yes (%)	No (%)
Lifetime traumatic event	59 (95.2)	3 (4.8)
Relational trauma *	54 (85.7)	
Non-relational trauma	9 (14.3)	
Traumatic event before AN onset	40 (76.9)	12 (23.1)
Multiple traumas	38 (62.3)	23 (37.7)
Current PTSD	14 (22.6)	48 (77.4)

PTSD = Post-traumatic Stress Disorder; * = childhood abuse and/or following LEC items: physical violence, being assaulted with a weapon, sexual assault, other kinds of sexual assault, being kidnapped.

**Table 4 medicina-57-00002-t004:** Differences in clinical presentation between young patients with AN with and without PTSD diagnosis.

Inpatients with AN (n = 64)					
	PTSD (n = 14)	Non-PTSD (n = 48)	Test Statistics		
	Mean (SD)	Mean (SD)	Z	*p*	*p* *
Duration of illness, years	4.1 (3.5)	3.3 (2.5)	−0.342	0.732	0.945
Age at onset, years	17.3 (3.4)	16.6 (2.1)	−1.119	0.263	0.436
BMI	16.4 (2.8)	14.8 (2.5)	−2.341	**0.019**	**0.012**
N of previous AN-related hospitalizations	2.4 (1.9)	1.9 (1.4)	−0.688	0.492	0.518
EDE-Q					
Restraint	3.5 (2.0)	2.5 (1.9)	−1.575	0.115	0.437
Food concern	3.5 (1.6)	2.8 (1.5)	−1.630	0.103	0.940
Shape concern	4.7 (1.2)	3.4 (1.6)	−2.233	**0.026**	0.117
Weight concern	4.2 (1.3)	2.9 (1.9)	−2.120	**0.034**	0.240
Total score	3.9 (1.4)	2.9 (1.6)	−2.093	**0.036**	0.291
BDI	20.4 (9.4)	15.6 (7.7)	−1.408	0.159	0.876
STAI-State	62.6 (10.4)	54.4 (12.6)	−1.799	0.072	0.487
STAI-Trait	61.1 (21.0)	55.6 (14.6)	−2.201	**0.028**	0.838
DES	35.8 (20.4)	18.2 (14.4)	−3.260	**0.001**	-

BMI = body mass index; EDE-Q = eating disorder examination questionnaire; BDI = Beck depression inventory; STAI = State-trait anxiety inventory; DES = dissociative experiences scale. *p* * = *p*-value after adjusting for DES score.

## Data Availability

Data are not publicly available, please refer to the corresponding author if needed.

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
