# Peer review of "Young Patients with Anorexia Nervosa: The Contribution of Post-Traumatic Stress Disorder and Traumatic Events"

_medicina, 2020, doi:10.3390/medicina57010002_

Round 1

Reviewer 1 Report

General comment:

The manuscript entitled “Young patients with anorexia nervosa: the contribution of Post-Traumatic Stress Disorder and traumatic events” describes an interesting study in a clinical sample with anorexia nervosa, outlining important concerns and implications for clinical practice. Nevertheless, there are quite a few concerns that diminished my enthusiasm for this manuscript. The Introduction section begins with a too broad overview of psychiatric disorders and risk factors, although the focus should be primarily on eating disorders. In that regard, the second part of the Introduction would benefit from the presentation of existing studies in this area rather than simply stating that studies are scarce. Rationale for the study could be scribed more extensively, and hypotheses are missing. Moreover, some parts of the manuscript, especially those related to presenting and discussing results, appear a bit disorganised. A clearer structure of presentation of findings would be beneficial, rather than just mentioning variables in which group differences were detected. Also, a clearer rationale for examining these specific relationships would be of use. In the current form, it seems that every potential difference was analysed, without a clear reason why is it important and how it contributes to the overall research question. Finally, the beginning statement as well as the ending of the Conclusion section are too strong (and too general) given there are indeed studies that investigated PTSD and risk factors in AN samples at this age. If, however, authors believe the existing literature does have a gap, then this should be supported with evidence clearly.

Specific comments:

L20: What does “body-related symptoms” indicate? Please amend.

L28: Abbreviations should be used consistently after first mention of “eating disorder” term. Please amend.

L38: What does “familiarity” mean in this context? Could authors please explain this term in the Introduction?

L77: Why only 25 years old or younger participants? Was this a convenience sample?

Could the authors report psychometric properties (such as internal consistency) of the administered questionnaires? DES scale was not described in the Methods section (or mentioned in the Abstract).

L124: What does “numerically inhomogeneous” mean here? That the assumption of homogeneity of variance was not met? Please clarify. Relatedly, authors say that “Wilcoxon-Mann-Whitney test was run to investigate differences between continuous variables” but the purpose of this test is to inspect differences between two groups. Same goes for the Fisher’s Exact test.

L126: Stating that ANCOVA was conducted to explore the role of potential confounders does not make much sense since its purpose is to control for the effect of confounding variables when examining differences between groups in DV - could this be reformulated please?

L131: Does ±2.3 in parenthesis indicate SD or range? Please specify. 

The category “traumatic event before AN onset” is unclear - aren’t all traumatic events assessed as a risk factor for AN development? Also, it would be interesting to see percentages for e.g. physical violence, sexual abuse, etc.

L150: Please paraphrase “reported significantly higher scores in BMI” as participants’ BMI was calculated based on measures taken by nurses.

Could the authors provide justification why dissociative symptoms were used as a covariate when examining comparing between participants with and without PTSD diagnosis?

L177-178: The first part of the sentence “trauma history did not differentiate..” seems at odds with the rest of the sentence. Isn’t higher body image disturbance part of clinical severity? Please amend.

L183: Stating that findings of the present study support genetic contribution in the etiology of eating disorders is not appropriate given this study did not examine genetic influences.

L185: Claims that something is a risk factor for EDs without using a prospective design, or comparing clinical and control groups at minimum, should be tentative.  

L217: Could the last part of the sentence (previous findings) be clarified?

L218-220: Does this statement refer to AN patients with PTSD or AN patients in general? Please specify.

L237-240: Study limitations should be mentioned prior to conclusion.

Minor concerns:

L60: Some expressions sound odd, for instance, “literature observed..”

L74: Restricter = restricting; binge-purging = binge-eating/purging 

L151: food concern scales = Eating Concern subscale

Author Response

As an overall statement we would like to thank the reviewers for their fruitful and thoughtful comments which we believe improved our manuscript.

REVIEWER #1

The manuscript entitled “Young patients with anorexia nervosa: the contribution of Post-Traumatic Stress Disorder and traumatic events” describes an interesting study in a clinical sample with anorexia nervosa, outlining important concerns and implications for clinical practice.

Thank you for these encouraging comments.

Nevertheless, there are quite a few concerns that diminished my enthusiasm for this manuscript. The Introduction section begins with a too broad overview of psychiatric disorders and risk factors, although the focus should be primarily on eating disorders. In that regard, the second part of the Introduction would benefit from the presentation of existing studies in this area rather than simply stating that studies are scarce. Rationale for the study could be scribed more extensively, and hypotheses are missing. Moreover, some parts of the manuscript, especially those related to presenting and discussing results, appear a bit disorganised. A clearer structure of presentation of findings would be beneficial, rather than just mentioning variables in which group differences were detected. Also, a clearer rationale for examining these specific relationships would be of use. In the current form, it seems that every potential difference was analysed, without a clear reason why is it important and how it contributes to the overall research question. Finally, the beginning statement as well as the ending of the Conclusion section are too strong (and too general) given there are indeed studies that investigated PTSD and risk factors in AN samples at this age. If, however, authors believe the existing literature does have a gap, then this should be supported with evidence clearly.

Thank you for raising these points. Overall, we agree with the reviewer about the introduction; however, this manuscript was focused on the special issue on risk factors in psychiatry, so we adapted the introduction accordingly. Nevertheless, following your suggestions, we tried to ameliorate the second part of the introduction, expanding the description of the existing studies (e.g., L60-L63: “Furthermore, childhood abuse was described as a non-specific risk factor for ED: in particular, the incidence of bulimic syndromes was estimated as 2.5 times higher in patients reporting an episode of childhood sexual abuse, with an increasing risk in case of multiple episodes (Sanci et al., 2008)”). Moreover, we added hypotheses (L85-88 “Concerning the first goal, we expected to find a higher frequency of certain (e.g., trauma-related) risk factors; secondly, we hypothesized to observe a more severe clinical presentation in patients with AN and comorbid PTSD when compared to those with AN but without PTSD”.), and we expand the rationale of the study as it follows: “Therefore, some gaps in literature need to be noted: few studies investigated substance and alcohol abuse as risk factor for AN as well as young patients at the onset of their illness; still, to the authors’ knowledge, very few studies focused on both familial and environmental risk factors. Moreover, Solmi et al., (2020) with an umbrella review accounted for a lack of well-established risk factors for EDs; in fact, even considering several risk factors (e.g., childhood sexual abuse, physical abuse, substance use, impulsivity), strong evidence was not found for any ED, particularly AN (Solmi et al., 2020)” (L70-L76).

Specific comments:

  1. L20: What does “body-related symptoms” indicate? Please amend.

       A definition of “body-related symptoms” was provided at L21-L22: “i.e., those symptoms        impacting on body image and perception and leading to body concerns”.

  1. L28: Abbreviations should be used consistently after first mention of “eating disorder” term. Please amend.

Thank you, done (L25).

  1. L38: What does “familiarity” mean in this context? Could authors please explain this term in the Introduction?

The definition of the term was provided in L39 as it follows: “i.e., presence of psychopathological symptoms in patients’ first or second-degree relatives”.

  1. L77: Why only 25 years old or younger participants? Was this a convenience sample?

This age range became inclusion criteria in order to meet the requirements of the special issue focusing on young patients.

  1. Could the authors report psychometric properties (such as internal consistency) of the administered questionnaires? DES scale was not described in the Methods section (or mentioned in the Abstract).

Thank you, we reported internal consistency for each questionnaire (except for LEC that is a simple list of events). DES was described in methods (L142-L147) and listed in the abstract (L17).

  1. L124: What does “numerically inhomogeneous” mean here? That the assumption of homogeneity of variance was not met? Please clarify. Relatedly, authors say that “Wilcoxon-Mann-Whitney test was run to investigate differences between continuous variables” but the purpose of this test is to inspect differences between two groups. Same goes for the Fisher’s Exact test.

We reworded these sentences in L151-L154 as it follows: “As the distribution of the groups was not normal, a Wilcoxon-Mann-Whitney test was run to investigate differences between groups as regards continuous variables. For differences between groups with categorical variables, Exact Fisher’s Test was used instead”.

  1. L126: Stating that ANCOVA was conducted to explore the role of potential confounders does not make much sense since its purpose is to control for the effect of confounding variables when examining differences between groups in DV - could this be reformulated please?

Thank you for the suggestion, the sentence was reformulated in L154: “Finally, analysis of covariance (ANCOVA) was conducted to control for possible confounding variables”.

  1. L131: Does ±2.3 in parenthesis indicate SD or range? Please specify.

It indicates the standard deviation, we clarified it (L160).

  1. The category “traumatic event beforeAN onset” is unclear - aren’t all traumatic events assessed as a risk factor for AN development? Also, it would be interesting to see percentages for e.g. physical violence, sexual abuse, etc.

We tried to make the category clearer explaining it in L79-L81: “focusing on childhood abuse, traumatic events (discriminating between events occurred during the life-span and those specifically occurred before the onset of AN, and between relational and non-relational traumas)”.  Moreover, we added the percentages of the traumatic events in the result section (L174-177): “In particular, concerning relational traumas, 37.7% of patients experienced physical violence, 8.2% were assaulted with a weapon, 9.8% suffered sexual assault and 23% other kind of sexual assault, and 4.9% reported to have been kidnapped”. 

  1. L150: Please paraphrase “reported significantly higher scores in BMI” as participants’ BMI was calculated based on measures taken by nurses.

Thank you for the comment, we paraphrased as it follows: “Patients with PTSD comorbidity had a higher BMI than non-PTSD patients. Furthermore, patients with a PTSD diagnosis reported significantly higher scores in shape concern” (L183).

  1. Could the authors provide justification why dissociative symptoms were used as a covariate when examining comparing between participants with and without PTSD diagnosis?

We decided to control for the role of dissociative symptoms since they are very frequent and important sequelae of relational trauma (Longo et al., 2020). We clarified this point in the text: “Since relational traumatic events lead often to dissociative symptoms (Longo et al., 2020), we controlled our findings for this covariate as well” (L186-L187).

  1. L177-178: The first part of the sentence “trauma history did not differentiate..” seems at odds with the rest of the sentence. Isn’t higher body image disturbance part of clinical severity? Please amend

The sentence was amended in L213-L216: “Second, overall clinical severity was not different between patients with and without trauma history, while patients with AN and comorbid PTSD showed more severe body-related symptoms”.

  1. L183: Stating that findings of the present study support genetic contribution in the etiology of eating disorders is not appropriate given this study did not examine genetic influences.

The sentence was reworded, now it sounds “Taken together, our results are in line with studies supporting the familial contribution to the etiology of AN (Bulik et al., 2007; Baker et al., 2018)” (L221)

  1. L185: Claims that something is a risk factor for EDs without using a prospective design, or comparing clinical and control groups at minimum, should be tentative. 

We agree with the comment, so the cross-sectional design of the study was described as a limitation of our study (L273).

  1. L217: Could the last part of the sentence (previous findings) be clarified?

The sentence was reformulated as it follows: “in line with previous literature describing a higher frequency of PTSD diagnosis in individuals predisposed to anxiety and stress-related psychopathology (Gilbertson et al., 2002)” (L253-L255)

  1. L218-220: Does this statement refer to AN patients with PTSD or AN patients in general? Please specify.

The statement refers to the loss of significance in differences between PTSD and non-PTSD groups after having controlled for dissociative symptoms, thus dissociation could play an important role in patients with AN and PTSD. The sentence was clarified (L256-258): “Therefore, it could be raised the hypothesis that the post-traumatic dissociation could be the main symptomatology cluster involved in the severity of eating-related symptoms in patients with AN and PTSD”

L237-240: Study limitations should be mentioned prior to conclusion.

Thank you for the suggestion, the paragraph was shifted at the end of discussion section, before the conclusion paragraph (L271-276)

Minor concerns:

L60: Some expressions sound odd, for instance, “literature observed..”

The expression was replaced now the sentence is “Moreover, literature reported shared genetic mechanisms between…” (L65)

L74: Restricter = restricting; binge-purging = binge-eating/purging 

The terms were modified as suggested although both terms are used in literature (L91)

L151: food concern scales = Eating Concern subscale

The term was corrected in L185

Reviewer 2 Report

The authors focused their study on analysis of risk factors in anorexia nervosa  (AN) patients, differences in clinical and eating-related symptoms between patients with and without a diagnosis of post-traumatic stress disorder (PTSD). Importance of the study is due to collection of data which possible were associated with hypothalamus-pituitary-adrenal (HPA) axis functioning.

Based on the authors results they have a right to conclude that to decrease  a severe AN form it is important to investigate the presence of risk factors and PTSD diagnosis among AN subjects. The results may suggest the need to address this character trait in therapeutic interventions.

Abstract: the structure of the abstract is correct and it contains the key information.

Introduction: it formulates  general hypothesis and the aim of the study as the first and second goals, however it would also clear to describe a character of  hypothesis O and 1

Methods: 

- medium number of participants, how a sample size was checked? any statistical data?

- any other clinical data of included inpatients from hospitalization period?

- is there a need of healthy control group to compare the results?

- exclusion criteria in terms of other psychiatric illnesses, inflammatory disorders,  endocrine disorders were not included, which is very important

Results: the results expressed  in four tables, where in Table 1 and 4 some values should be reduced to contain one decimal place only, there is no need to extend numbers

Discussion – supported by the well-chosen citations, in order to better understand the interest of these results,  please add a paragraph which stress  markers could be analysed among ANs ? cortisol only?

Conclusion: It is most welcome limitations of the study that you have include to this paragraph. I agree that examined risk factors like childhood trauma, familiarity of psychiatric disorders and co-diagnosis PTSD and AN can results in severity of clinical symptoms.

References: correct and enough positions

Author Response

As an overall statement we would like to thank the reviewers for their fruitful and thoughtful comments which we believe improved our manuscript.

REVIEWER # 2

The authors focused their study on analysis of risk factors in anorexia nervosa  (AN) patients, differences in clinical and eating-related symptoms between patients with and without a diagnosis of post-traumatic stress disorder (PTSD). Importance of the study is due to collection of data which possible were associated with hypothalamus-pituitary-adrenal (HPA) axis functioning.

Based on the authors results they have a right to conclude that to decrease  a severe AN form it is important to investigate the presence of risk factors and PTSD diagnosis among AN subjects. The results may suggest the need to address this character trait in therapeutic interventions.

Abstract: the structure of the abstract is correct and it contains the key information.

  1. Introduction: it formulates  general hypothesis and the aim of the study as the first and second goals, however it would also clear to describe a character of  hypothesis O and 1

Thank you, also reviewer #1 raised this point; we clarified our hypotheses at the end of introduction, L85-L88: “Concerning the first goal, we expected to find a higher frequency of certain (e.g., trauma-related) risk factors; secondly, we hypothesized to observe a more severe clinical presentation in patients with AN and comorbid PTSD when compared to those with AN but without PTSD”.

Methods: 

-2.  medium number of participants, how a sample size was checked? any statistical data?

Unfortunately, no power analysis was conducted.

- 3. any other clinical data of included inpatients from hospitalization period?

We agree with the referee’s suggestion; however, the study was designed for a special issue on risk factors, so we narrowed the focus on these variables and we decided not to added other data also following the comment offered by reviewer #1.

- 4. is there a need of healthy control group to compare the results?

We added the lack of a healthy control group to the limitations (L275-L276)

-5.  exclusion criteria in terms of other psychiatric illnesses, inflammatory disorders,  endocrine disorders were not included, which is very important

Thank you for this comment, we excluded patients with psychotic and bipolar disorders and substance abuse but we missed to specify these criteria. We have now added these important pieces of information to the paper – please see L98-L99: “(d) comorbid psychotic spectrum disorders and/or bipolar disorders; (e) substance and/or alcohol abuse”; other comorbid symptoms (e.g., depression and anxiety) normally are not exclusion criteria since they are deeply entangled with eating-related pathology.

  1. Results: the results expressed  in four tables, where in Table 1 and 4 some values should be reduced to contain one decimal place only, there is no need to extend numbers

Thank you we modified Tables 1 and 4.

  1. Discussion – supported by the well-chosen citations, in order to better understand the interest of these results,  please add a paragraph which stress  markers could be analysed among ANs ? cortisol only?

Thank you for the suggestion, we added this topic in L287-289: “In this context, future studies could focus on biological mechanism related to PTSD such as cortisol reactivity, already known to be altered in patients with AN (Monteleone et al., 2020), and epigenetic regulation (Howie et al., 2029). 

Conclusion: It is most welcome limitations of the study that you have include to this paragraph. I agree that examined risk factors like childhood trauma, familiarity of psychiatric disorders and co-diagnosis PTSD and AN can results in severity of clinical symptoms.

Thank you.
